# A Functional Network Driven by MicroRNA-125a Regulates Monocyte Trafficking in Acute Inflammation

**DOI:** 10.3390/ijms231810684

**Published:** 2022-09-14

**Authors:** Stephanie Tomasi, Lei Li, Ludwig Christian Hinske, Roland Tomasi, Martina Amini, Gabriele Strauß, Martin Bernhard Müller, Simon Hirschberger, Sven Peterss, David Effinger, Kristin Pogoda, Simone Kreth, Max Hübner

**Affiliations:** 1Department of Transfusion Medicine, Cell Therapeutics and Haemostaseology, LMU University Hospital, Ludwig Maximilians University München (LMU), 81377 Munich, Germany; 2Walter Brendel Center of Experimental Medicine (WBex), Ludwig Maximilians University München (LMU), 81377 Munich, Germany; 3Department of Anaesthesiology and Intensive Care Medicine, Research Unit Molecular Medicine, LMU University Hospital, Ludwig Maximilians University München (LMU), 81377 Munich, Germany; 4Institute for Digital Medicine, University Hospital Augsburg, Stenglinstrasse 2, 86156 Augsburg, Germany; 5Department of Anesthesiology, Fondazione IRCCS Istituto Nazionale dei Tumori, Via Venezian 1, 20133 Milano, Italy; 6Department of Cardiac Surgery, University Hospital, Ludwig Maximilians University München (LMU), 81377 Munich, Germany; 7Physiology, Institute for Theoretical Medicine, University of Augsburg, 86159 Augsburg, Germany

**Keywords:** microRNA, inflammation, monocyte trafficking, adhesion, chemotaxis

## Abstract

During the onset of acute inflammation, rapid trafficking of leukocytes is essential to mount appropriate immune responses towards an inflammatory insult. Monocytes are especially indispensable for counteracting the inflammatory stimulus, neutralising the noxa and reconstituting tissue homeostasis. Thus, monocyte trafficking to the inflammatory sites needs to be precisely orchestrated. In this study, we identify a regulatory network driven by miR-125a that affects monocyte adhesion and chemotaxis by the direct targeting of two adhesion molecules, i.e., junction adhesion molecule A (JAM-A), junction adhesion molecule-like (JAM-L) and the chemotaxis-mediating chemokine receptor CCR2. By investigating monocytes isolated from patients undergoing cardiac surgery, we found that acute yet sterile inflammation reduces miR-125a levels, concomitantly enhancing the expression of JAM-A, JAM-L and CCR2. In contrast, TLR-4-specific stimulation with the pathogen-associated molecular pattern (PAMP) LPS, usually present within the perivascular inflamed area, resulted in dramatically induced levels of miR-125a with concomitant repression of JAM-A, JAM-L and CCR2 as early as 3.5 h. Our study identifies miR-125a as an important regulator of monocyte trafficking and shows that the phenotype of human monocytes is strongly influenced by this miRNA, depending on the type of inflammatory stimulus.

## 1. Introduction

Acute inflammation paves the way for increased immune cell trafficking to the inflammatory focus. Initially, circulating neutrophils migrate rapidly to the inflamed tissue and create a proinflammatory microenvironment that promotes monocyte emigration from the bloodstream into the inflammatory area. This recruitment of circulating monocytes is indispensable for all phases of the inflammatory reaction, both the acute phase during which the inflammatory noxa is neutralised and also the resolution phase that eventually leads to the reconstitution of homeostasis [1,2]. Dysregulation of monocyte trafficking, however, disturbs this delicate balance and renders the immune system unable to resolve the inflammatory reaction [3]. This may result in persistent chronic inflammation with excessive tissue damage and triggers inflammation-associated diseases such as autoimmune disorders, atherosclerosis or cancer [4,5,6]. The mechanisms that orchestrate the complex network of monocyte trafficking, however, are currently not completely characterised.

Very recently, we could identify an miR-125a-driven network in endothelial cells consisting of at least five target genes regulating endothelial permeability during acute inflammation. In this study, we additionally revealed that miR-125a dramatically impairs monocyte migration by targeting the central receptor for monocyte chemotaxis, the C-C chemokine receptor type 2 (CCR2) [7]. It is more than likely that this cell-crossing functional synergism of miR-125a extends beyond the sole regulation of the CCR2 in monocytes. The current study was conducted to investigate whether miR-125a also represents a central hub in the regulation of monocyte function.

We here provide evidence that two sequential processes of monocyte trafficking, endothelial adhesion and chemotaxis, are modulated by miR-125a and via its novel targets, the transmembrane glycoproteins JAM-A, JAM-L and CCR2, respectively. The regulatory effect of miR-125a on all three targets could be verified in two different acute inflammatory settings; however, the direction of regulation was disparate depending on the inflammatory stimulus: sterile inflammation, evoked by cardiac surgery induced a repression of miR-125a with subsequent upregulation of target genes, resulting in increased monocyte trafficking, while LPS incubation mimicking bacterial infection led to opposite findings.

These data provide valuable new insights into the delicate regulation of monocyte trafficking during acute inflammation and underline the translational relevance of the miR-125a regulatory network, which is capable of either enhancing or impairing monocyte trafficking, depending on the inflammatory stimulus.

## 2. Results

### 2.1. miR-125a Modulates the Expression of Genes Involved in Monocyte Adhesion

To identify genes and pathways that are possibly regulated by miR-125a, a whole transcriptome microarray of monocytes transfected with miR-125a or a scrambled control was carried out (Figure 1A). By applying KEGG and STRING pathway analysis, we identified three signalling pathways (chemokine signalling, Wnt signalling and Rap1 signalling) that were disproportionately affected by overexpression of miR-125a. Importantly, these pathways are well-known regulators of leukocyte trafficking and particularly affect monocyte adhesion and chemotaxis [8,9,10,11,12,13]. Consequently, we combined these array results and pathway analyses with in silico target prediction algorithms TargetScan 7.2 (www.targetscan.org (last accessed on 15 July 2021) and miRIAD (http://bmi.ana.med.uni-muenchen.de/miriad/ (last accessed on 10 June 2021) to identify genes that (a) are highly expressed in monocytes, (b) directly mediate monocyte adhesion, (c) are significantly repressed after transfection of miR125a and (d) harbour putative binding sites of miR-125a in the 3′UTR. In addition to the already evaluated CCR2, this approach prompted us to investigate junction adhesion molecule A (JAM-A) and junction adhesion molecule-like (JAM-L) in a more detailed manner. Validation of the array results by qRT-PCR and flow cytometry confirmed significant repression of both JAM-A and JAM-L (JAM-A: −41.7% ± 2.7%, n = 7, *p* < 0.05; JAM-L: −45.3% ± 4.0%, n = 5, *p* < 0.01; Figure 1B–E).

### 2.2. JAM-A and JAM-L Are Direct Targets of miR-125a

To investigate whether miR-125a also impairs JAM-A and JAM-L levels via direct targeting, we first conducted in silico analyses and, thus, identified potential binding sites in the 3′UTR of both JAM-A and JAM-L. To provide experimental proof that both genes are bona fide targets of miR-125a, we next designed luciferase reporter plasmids containing the 3′UTRs of either JAM-A or JAM-L. The localisation of the putative binding sites of miR-125a in the respective 3′UTRs is indicated in Figure 2A. Cotransfection of these luciferase reporter constructs and miR-125a diminished luciferase activity of both JAM-A and JAM-L reporter constructs compared with scrambled controls (Figure 2B; JAM-A: −44.7% ± 1.9%, n = 6, *p* < 0.05; JAM-L: −26.8% ± 9.8%, n = 5, *p* < 0.05). These findings confirm the downregulation of JAM-A and JAM-L via direct interaction of miR-125a with their respective 3′UTRs.

### 2.3. miR-125a Impairs Monocyte Adhesion to Inflammatory Endothelial Cells by Targeting JAM-A and JAM-L

We next assessed whether overexpression of miR-125a indeed impacts monocyte recruitment to endothelial cells during acute inflammation. To this end, human monocytes were perfused across TNF-activated human umbilical vein endothelial cells (HUVECs). Intriguingly, we were able to show that overexpression of miR-125a resulted in a significant reduction in monocyte adhesion to endothelial cells as measured by calcein fluorescence (Figure 3A; relative fluorescence activity: −34.1% ± 4.7%, n = 8, *p* < 0.01).

Subsequently, RNA interference was used to evaluate whether the newly identified direct targets of miR-125a, JAM-A and JAM-L indeed account for the observed phenotypic alteration and performed RNA interference experiments (Appendix A; knockdown efficiency: CCR2:−63.8% ± 7.7%, n = 3, *p* < 0.01; JAM-A: −74.0% ± 3.7%, n = 3, *p* < 0.05; JAM-L: −67.8% ± 4.1%, n = 3, *p* < 0.01) In line with previous studies [14,15,16,17,18], monocyte adhesion to endothelial cells was significantly reduced upon JAM-A and JAM-L knockdown (Figure 3B,C; JAM-A: reduction by 25.4% ± 5.3%, n = 6, *p* < 0.05; JAM-L: reduction by 38.4% ± 11.2%, n = 4, *p* < 0.05). CCR2 knockdown, however, did not inhibit monocyte adhesion to endothelial cells. These results suggest that JAM-A and JAM-L regulate monocyte adhesion to endothelial cells during acute inflammation.

### 2.4. Expression of CCR2, JAM-L, JAM-A and miR-125a during Inflammation

In homeostatic conditions, monocyte trafficking to the perivascular area is in a steady state, replacing tissue-resident, senescent macrophages. This physiological turnover is also orchestrated by adhesion molecules such as receptors of the JAM family, CCRs and integrins [18]. Upon an inflammatory insult, however, early immune cell trafficking to the inflammatory focus is essential to initiate the resolution phase and eventually reconstitute homeostasis [18,19,20]. Previously, we were able to show that stimulation of monocytes with the proinflammatory cytokine IL-6 significantly enhances CCR2 expression [7]. Here, we pursued a more clinical model of acute inflammation: cardiac surgery with the use of cardiopulmonary bypass (CPB) mounts a strong yet sterile and clearly defined systemic inflammatory response and, thus, represents an excellent model to investigate acute systemic inflammation ex vivo. To this end, we quantified miR-125a, CCR2- JAM-L and JAM-A levels in monocytes isolated from patients before CPB (T1) and on the first postoperative day (T3). As compared with T1, the systemic inflammatory response at T3 led to decreased levels of miR-125a, whereas expression of CCR2, JAM-L and JAM-A was significantly induced (Figure 4A,B; miR-125a: −39.8% ± 13.6%, n = 15, *p* < 0.05; CCR2: induction 2-fold ± 0.1, n = 15, *p* < 0.001; JAM-L: induction 1.1-fold ± 0.05, n = 15, *p* < 0.05; JAM-A: induction 1.17-fold ± 0.06, n = 15, *p* = 0.002). It can thus be assumed that monocyte trafficking to the perivascular area is enhanced by miR-125a downregulation during the onset of acute inflammation.

However, once monocytes have egressed from the bloodstream and entered the inflammatory area, the milieu dramatically changes: high amounts of damage- or pathogen-associated molecular patterns (DAMPs/PAMPs) such as LPS lead to toll-like receptor (TLR) 4-specific monocyte activation, induce monocyte-to-macrophage differentiation and induce a stationary monocyte phenotype [8,21,22]. These processes are indispensable to efficiently initiate the resolution of inflammation [23,24,25]. We recently showed that monocyte-to-macrophage differentiation indeed alters miR-125a and CCR2 levels and that overexpression of miR-125a inhibits monocyte chemotaxis [7]. Of note, previous studies provided evidence that the novel miR-125a targets JAM-A and JAM-L also impact immune cell migration and chemotaxis [26,27,28,29]. We, therefore, hypothesised that TLR-4 activation also modulates miR-125a and, consequently, levels of all three targets of miR-125a, that is, CCR2, JAM-A and JAM-L, thus contributing to a stationary monocyte phenotype. Incubation of monocytes with the TLR-4 activator LPS induced a decrease in CCR2, JAM-A and JAM-L expression as early as 3.5 h after administration (Figure 4C; CCR2: −93.1% ± 2.2%, n = 5, *p* < 0.001; JAM-A: −68.4% ± 2.5%, n = 5, *p* < 0.001; JAM-L: −23.2% ± 8.2%, n = 4; *p* < 0.05). Concomitantly, miR-125a levels were increased after TLR-4 activation (Figure 4D, 3.5 h after administration, +31.4% ± 8.2%, n = 4, *p* < 0.05; Figure 4E, 20 h after administration, induction 5.67-fold ± 1.86, n = 5, *p* < 0.01).

To investigate whether these changes indeed impact monocyte chemotaxis, we next assessed CCR2-specific monocyte migration via time-lapse microscopy. This revealed a dramatic reduction in directed migration, velocity and accumulated distance (Figure 5).

These results provide evidence that miR-125a, at least in part, confers sedentariness within the inflammatory area after TLR4-activation.

## 3. Discussion

Inflammatory insults, such as infections or trauma, trigger the secretion of specific proinflammatory cytokines, provoking increased trafficking of leukocytes to the inflammatory focus [30,31]. This process is a multistep sequence involving receptor-mediated interactions between leukocytes and endothelial cells. After firm adhesion to endothelial cells, leukocytes eventually transmigrate through the endothelial monolayer and chemotactically migrate towards the inflammatory site [32,33].

These interactions between endothelial cells and leukocytes constitute a complex functional network that needs to be precisely orchestrated to mount fast yet appropriate immune responses that eventually reconstitute homeostasis.

Recently, we were able to identify miR-125a as a central signalling hub regulating endothelial permeability by direct targeting multiple genes. Within the same study, we showed that miR-125a also modulates monocyte migration and chemotaxis by directly targeting C-C chemokine receptor type 2 (CCR2), the central receptor for monocyte chemotaxis. It is a widely accepted phenomenon that miRs target several functionally interrelated genes simultaneously and, thus, strongly increase their regulatory potential [34,35]. We thus assumed that miR-125a also interacts with additional genes, exceeding the sole targeting of CCR2 and, thus, creating a functional network that regulates monocyte trafficking.

To identify potential targets of miR-125a that possibly affect monocyte trafficking, we transfected primary human monocytes with miR-125a and analysed gene expression by whole transcriptome microarrays. In silico analyses revealed that the pathways *GO_cell–cell adhesion* and *GO_leukocyte cell–cell adhesion* contain a disproportionate number of genes repressed by miR-125a, supporting our assumption that miR-125a is involved in monocyte trafficking. To assess whether miR-125a indeed alters monocyte recruitment to endothelial cells, we conducted adhesion assays under flow conditions and found that transfection of miR-125a indeed reduced the number of monocytes adhering to endothelial cells. To identify direct target genes of miR-125a that account for the observed phenotypic changes, we extracted genes from the microarray dataset that are (a) involved in the regulation of cell adhesion, (b) significantly regulated by miR125a and (c) strongly expressed in monocytes. From the resulting candidate genes, we identified two junctional adhesion molecules (JAMs), JAM-A and JAM-L, as novel direct targets of miR-125a. Previous studies revealed that the JAM family is specifically important during monocyte extravasation, migration and chemotaxis and that both genes are expressed in monocytes [27,29,36,37,38]. We corroborated the impact of JAM-A and -L on monocyte trafficking by RNA interference experiments, revealing that individual knockdown of either JAM-A or -L strongly hampered monocyte adhesion to endothelial cells. As expected, knockdown of the known miR-125a-target CCR2 had no effect on cellular adhesion. These findings reveal that miR-125a not only impacts monocyte migration via regulating CCR2 but also strongly influences monocyte adhesion by targeting the expression of JAM-A and -L.

We previously showed that miR-125a levels in monocytes are regulated by proinflammatory stimulation with IL-6 [7]. Severe inflammation, however, releases a plethora of cytokines, potentially revoking an artificial in vitro effect. Thus, to depict the changes to circulating inflammatory monocytes in a translational setting, we investigated monocytes from patients undergoing cardiac surgery with the use of CPB since this procedure mounts a strong, sterile and reproducible acute inflammatory reaction. [39]. This approach clearly showed that acute inflammation reduces levels of miR-125a and enhances the expression of its target genes, JAM-A, JAM-L and CCR2, concomitantly. These results suggest a regulatory impact of miR-125a on the egress of circulating monocytes to the perivascular area during acute inflammation in vivo.

After entering an inflamed perivascular compartment, however, monocytes are challenged with high concentrations of PAMPs and DAMPs, for example, LPS, which activate monocytes in a TLR-specific manner [40]. Within the perivascular inflammatory area, activated monocytes become stationary and differentiate into macrophages to efficiently combat the inflammatory stimulus and resolve inflammation [41]. The underlying mechanisms that confer sedentariness, however, are currently not completely clear. We here provide novel evidence that miR-125a is dramatically induced as early as 3.5 h after TLR-specific stimulation with LPS. Concomitantly, JAM-A, JAM-L and CCR2 levels were strongly repressed, and, as expected, time-lapse microscopy revealed a dramatic repression of monocyte mobility after LPS contact. This clear contrast to activated circulating monocytes suggests that miR-125a considerably contributes to the modulation of their target gene’s expression levels during acute inflammation. It is tempting to speculate that both the type of inflammatory stimulation—nonspecific v. TLR-specific—and the inflammatory compartment dictates monocyte gene expression and, eventually, trafficking.

In summary, we propose that miR-125a is a previously underestimated regulator of monocyte trafficking, targeting three functionally interrelated genes during the early onset of inflammation. While previous studies revealed a regulatory role of miR-125a during monocyte-to-macrophage differentiation [7,42], we here characterised a disparate regulation of miR-125a a) in circulating monocytes during the early onset of inflammation and b) within the inflammatory microenvironment after contact with PAMPs/DAMPs.

Taken together, these results suggest that the regulatory network around miR-125a is a novel central mechanism for the modulation of different processes during monocyte trafficking, that is, adhesion to endothelial cells and chemotaxis. This miR-125a-mediated functional synergism that is preserved during acute inflammation may open up new roads in the diagnosis and therapy of inflammation-driven diseases.

## 4. Materials and Methods

### 4.1. PBMC Isolation and Microbead-Based Monocyte Extraction

Peripheral blood mononuclear cells (PBMCs) were isolated from whole blood of healthy donors by Ficoll density gradient centrifugation using Histopaque-1077 (Sigma-Aldrich, St. Louis, MO, USA) according to the manufacturer’s instructions. PBMCs were counted using a Vi-Cell XR automated cell counter (Beckman-Coulter, Brea, CA, USA). Pan monocytes composed of classical (CD14^++^ CD16^−^), nonclassical (CD16^++^ CD14^+^) and intermediate monocytes (CD16^+^ CD14^++^) were isolated from PBMCs using the Pan Monocyte Isolation Kit, human (Miltenyi Biotec, Germany, Cat. No. 130-096-537), on an AutoMACS separation device (Miltenyi Biotec), according to the manufacturer’s instructions.

### 4.2. Transfection of Primary Human Monocytes

Primary human monocytes were transfected with 50 nM Ambion hsa-miR125 pre-miR miRNA (Thermo Fisher Scientific, Waltham, MA, USA) or ON-TARGET SMARTpool siRNA (Dharmacon, Lafayette, CO, USA) targeting the gene of interest. Transfection was performed by electroporation using the NEON transfection system (1 pulse, 30 ms, 1900 V). Transfected cells were incubated at 37 °C and 5% CO_2_ in antibiotics-free RPMI (Sigma-Aldrich, Schnelldorf, Germany). Transfection efficiency was determined by flow cytometry using Cy3 dye-labelled pre-microRNA as a negative control (Thermo Fisher Scientific). Transfection efficiency was always above 60%.

### 4.3. HUVEC Isolation and Cell Culture

Primary human umbilical vein endothelial cells (HUVECs) were isolated from umbilical cords of healthy neonates directly after caesarean delivery at the Department of Gynecology and Obstetrics, University Hospital, LMU Munich, Germany. Written informed consent was obtained from the mothers prior to the donation. HUVECs were isolated from the umbilical vein vascular wall by collagenase A (Roche, Penzberg, Germany) treatment and cultured in Endothelial Cell Growth Medium (ECGM; PromoCell, Heidelberg, Germany) with 10% FCS (Biochrom AG, Berlin, Germany) and 1% penicillin/streptomycin (pen/strep; Gibco). HUVECs used for experiments were not cultured beyond the fourth passage.

### 4.4. DNA and RNA Extraction

Genomic DNA was extracted from 80 μL of heparinised whole blood of healthy volunteers using the QIAamp DNA Mini Kit according to the manufacturer’s instructions (Qiagen, Hilden, Germany). Whole blood samples from septic patients and healthy volunteers were obtained using the PAXgene Blood RNA System (Qiagen). Total RNA was extracted from cell lysates using the RNAqueous Isolation Kit (Ambion, Waltham, MA, USA) according to the manufacturer’s instructions. To quantify miR-125a expression, total RNA was extracted with the mirVana miRNA Isolation Kit (Ambion, Thermo Fisher Scientific, Austin, TX, USA). After RNA isolation, the TURBO DNA-free Kit (Invitrogen, Waltham, MA, USA) was used to remove DNA contamination. RNA amount and quality were assessed using a NanoDrop 2000 spectrophotometer (Thermo Fisher).

### 4.5. cDNA Synthesis and Quantitative Real-Time PCR

cDNA was synthesised using Oligo-dT primers, random hexamers (Qiagen, Venlo, The Netherlands), dNTPs, RNAse OUT, and Superscript III reverse transcriptase (Invitrogen, Waltham, MA, USA). Quantitative polymerase chain reaction (qPCR) was performed using a LightCycler480 (Roche Diagnostics, Penzberg, Germany) as previously described [43]. For PBMC, SDHA and TBP were used as reference genes. For monocytes, B2M and TBP served as reference genes. For quantification of miR-125a-5p by qRT-PCR, the TaqMan MicroRNA Assay (Applied Biosystems, Assay ID: 002198) was used according to the manufacturer’s instructions. U47 (Assay ID 001223, Applied Biosystems, Waltham, MA, USA) served as the reference gene. Taqman PCR conditions comprised initial denaturation for 10 min at 95 °C and 50 cycles of 95 °C for 15 s, 60 °C for 60 s and 40 °C for 30 s.

### 4.6. Microarray

Expression analysis of the entire transcriptome of transfected monocytes was carried out using the Affymetrix GeneChip Primeview Human Gene Expression Array (IMGM Laboratories GmbH, Martinsried). Bioinformatics was performed by the Anesthesia and Critical Care Informatics and Data Analysis Group (Department of Anesthesiology, University Hospital LMU Munich). Microarray data are available on ArrayExpress under the accession number MTAB-11824. In silico identification of potential miRNA binding sites was performed using the public databases TargetScan (www.targetscan.org) [44] and MiRIAD (https://www.miriad-database.org/) [45].

### 4.7. Cloning of Reporter Constructs

The 3′UTRs of JAM-A and JAM-L were amplified from genomic DNA by PCR. The primer sequences are listed in Appendix A. PCR products were ligated into the StrataClone Blunt Vector pSC-B-amp/kan (Agilent Technologies) according to the manufacturer’s instructions and then subcloned into the psiCHECK-2 vector (Promega, Mannheim, Germany) using XhoI/PmeI restriction enzymes (New England Biolabs) and T4 DNA ligase (Roche Diagnostic GmbH, Mannheim, Germany). Site-directed mutagenesis of plasmid DNA was conducted using specific primers (Metabion) and the QuikChange Lightning Site-Directed Mutagenesis Kit (Agilent Technologies) according to the manufacturer’s protocol. Plasmids were purified using the Pure Yield Plasmid Midiprep System (Promega). The correct sequence and orientation of the 3’UTR in the vector were verified by Sanger sequencing (Eurofins Operon, Ebersberg, Germany).

### 4.8. Reporter Gene Assays

HEK-293 cells were purchased from the American Type Culture Collection and cultured in Dulbecco’s Modified Eagle Medium (Gibco) supplied with 10% FCS, 2% L-glutamine, 1% penicillin/streptomycin and 1% MEM NEAA at 37 °C and 5% CO_2_. HEK-293 cells used for experiments were not cultured beyond the 20th passage. Cotransfection of luciferase reporter plasmids and pre-miR-125a was carried out using 100,000 HEK-293 cells and 1 μg of psiCHECK-2 plasmid containing either the 3′UTR of JAM-A or JAM-L, respectively. Transfections were conducted using the NEON electroporation device (Thermo Fisher, 1150V, 2 pulses, 20 msec). After 40 h, cells were harvested, and reporter gene activity was assessed using the Dual-Glo Luciferase Assay system (Promega), according to the manufacturer’s instructions, on a microplate reader (FilterMax F3, Molecular Devices). All experiments were performed in triplicate.

### 4.9. Flow Cytometry

All flow cytometric analyses were performed on an Attune Acoustic Focusing Cytometer (Thermo Fisher). Monocytes were harvested and resuspended in PBS containing 1% BSA. Fc receptor blocking was carried out via TruStain FcX receptor blocker. Cells were stained with fluorophore-conjugated antibodies raised against JAM-A (Biolegend, San Diego, CA, USA, Cat. No. 353503, conjugation: FITC) or JAM-L (R&D, USA, Cat. No. FAB34491P, conjugation: PE). The fluorescence signal was detected using a 488 nm laser and a 530 nm/30 nm emission filter. Data were analysed by FlowJo software, version 10 (FlowJo, Ashland, Catlettsburg, KY, USA).

### 4.10. Adhesion Assays

For optimal conditions, µ-Slides I 0.6 Luer (ibidi) were precoated with 50 μg/mL Poly-L-lysine for 20 min, followed by 0.5% glutaraldehyde for 15 min and 0.2% gelatin for 10 min at room temperature. Between each step, slides were washed three times with PBS. The slides were then filled with ECGM and incubated overnight at 37 °C and 5% CO_2_. Next, HUVECs (1 × 10^6^ cells/mL) were seeded in µ-Slides, incubated for 3 h at 37 °C and 5% CO_2_, and then connected to the ibidi pump system, applying shear stress of 3 dyn/cm^2^ for 3 h. Then, shear stress was increased to 5 dyn/cm^2^ for 12 h, and HUVECs were then activated with 2.5 ng/mL TNF for 6 h. Transfected monocytes were activated with 25 ng/mL CCL2 for 45 min, and then subsequently stained with 1 µM calcein-AM for 30 min and washed twice with PBS to remove excess calcein. The µ-Slide was connected to a new pump system containing fresh ECGM. Monocytes (3 × 10^6^) were added to the pump reservoir and perfused across the HUVEC monolayer for 45 min at 5 dyn/cm^2^. Thereafter, perfusion was increased to 10 dyn/cm^2^ for 5 min and to 20 dyn/cm^2^ for 10 min in order to remove nonadherent monocytes from the endothelium. The slides were separated from the pump system and washed six times with 150 μL PBS. Subsequently, three-phase contrast and fluorescent microscopy images were obtained from each slide. Finally, all slides were treated with equal amounts of trypsin and washed with equal amounts of PBS. Subsequently, fluorescence was analysed using a fluorescent reader (FilterMax F3; excitation, 485 nm; emission, 535 nm).

### 4.11. Chemotaxis Assay

Chemotaxis was assessed as previously described [7,46]. Briefly, chemotaxis slides (ibidi) and RPMI media were equilibrated at 37 °C and 5% CO2 in a humidified incubator one day prior to the experiment. On the day of the experiment, 2 × 10^6^ monocytes were resuspended in 175 µL RPMI supplemented with 20% FCS, 1% penicillin/streptomycin/glutamine and 1% HEPES. After adding 25 µL of bovine collagen I (Thermo Fisher), 6 µL was introduced in the µ-Slide chemotaxis (ibidi). The slide was placed in a cell culture incubator for 1 h to ensure proper ECM matrix formation. Immediately before time-lapse microscopy, the reservoirs were loaded with RPMI media alone or RPMI media supplemented with 50 ng/mL CCL2, respectively, to create a chemokine concentration gradient. Time-lapse microscopy was performed on a Zeiss Axio Observer Z1 equipped with a gas incubation system (ibidi). For each channel, three representative fields of vision were chosen, and pictures were obtained for 4 h (one picture/minute). Single-cell tracking analysis was performed using ImageJ and the Chemotaxis and Migration Tool (ibidi). At least 20 cells per sample were tracked.

### 4.12. Ethics

All subjects gave their written informed consent before they participated in the study. The study was conducted in accordance with the Declaration of Helsinki, and the protocol was approved by the ethics committee of the Ludwig-Maximilians-University Munich (LMU), project no. 17-491.

### 4.13. Statistics

Statistical evaluations were carried out using Graph Pad Prism 9 (GraphPad Software, Inc, USA). All datasets were tested for Gaussian distribution, and *p*-values were calculated using Student’s *t*-test or Mann–Whitney U test. All data are presented as mean ± SEM unless stated otherwise. *p* < 0.05 was considered statistically significant (* *p* < 0.05, ** *p* < 0.01, *** *p* < 0.001, n.s. = not significant). All experiments were performed in triplicate and repeated at least three times.

## Figures and Tables

**Figure 1 ijms-23-10684-f001:**
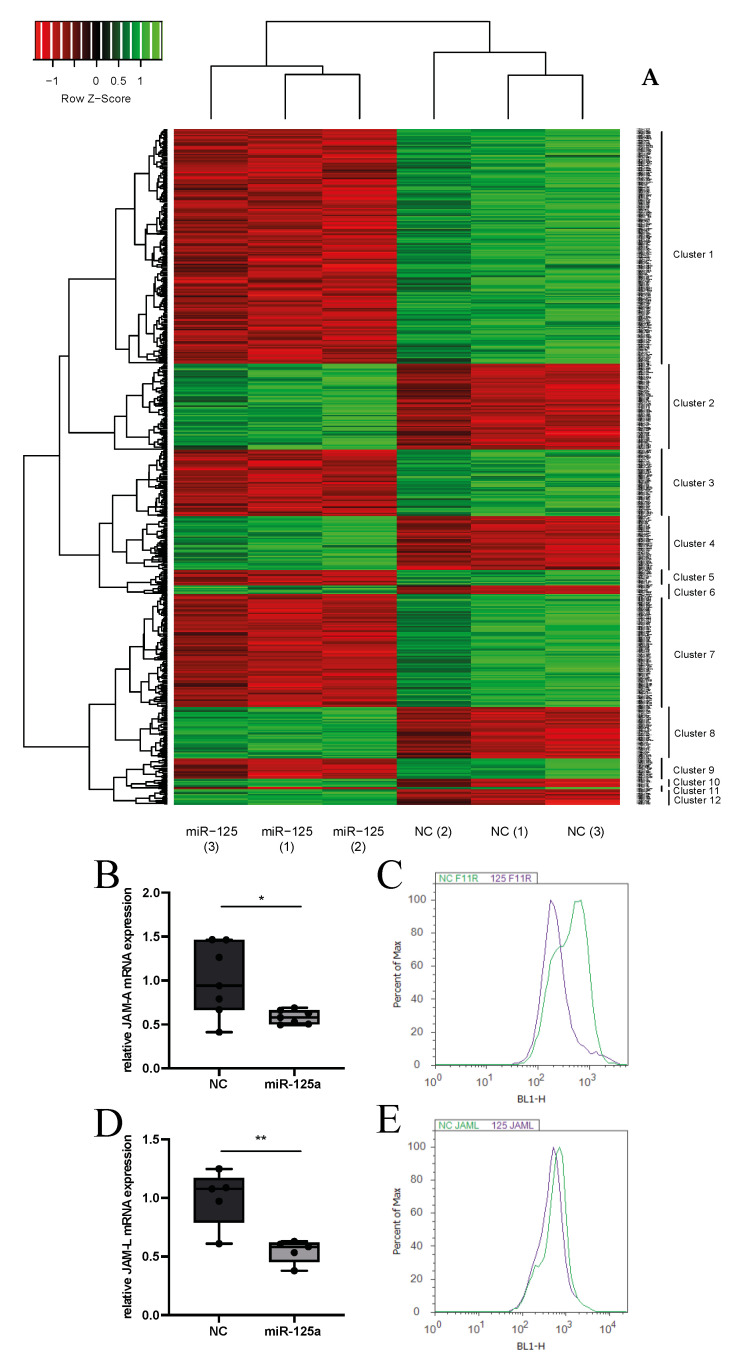
**miR-125a regulates the expression of genes involved in monocyte adhesion.** (**A**) Heatmap of differentially expressed genes in monocytes transfected with either miR-125a or scrambled control (NC) of three independent experiments and three individual donors, *p* < 0.05). Enhanced expression is indicated in green colour; reduced expression is indicated in red colour. The exact gene names, ordered by clusters, are provided in the Appendix A. mRNA expression of JAM-A (**B**) (n = 7, * *p* < 0.05) and JAM-L (**D**) (n = 5, ** *p* < 0.01) as measured by qRT-PCR (**C**,**E**). Surface protein expression of JAM-A and JAM-L in primary human monocytes after transfection of miR-125a (violet) or scrambled control (NC, green) as analysed by flow cytometry. A representative flow cytometric staining from a total of five independent experiments is shown.

**Figure 2 ijms-23-10684-f002:**
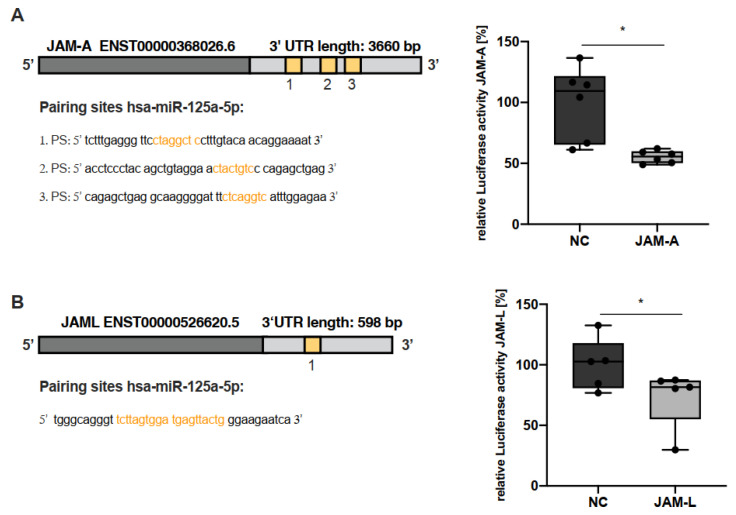
**miR-125a directly interacts with the 3′UTRs of JAM-A and JAM-L.** (**A**) Localisation of the three in silico-predicted miR-125a binding sites within the 3′UTR of JAM-A. The relative JAM-A 3′UTR luciferase reporter gene activity after cotransfection of the luciferase constructs with miR or scrambled control (NC) (n = 6, * *p* < 0.05). (**B**) Schematic representation of the in silico-predicted miR-125a binding site within the JAM-L 3′UTR. JAM-L 3′UTR luciferase reporter gene activity after cotransfection of the luciferase constructs with miR-125a or scrambled control (NC), normalised to firefly luciferase (n = 5, * *p* < 0.05).

**Figure 3 ijms-23-10684-f003:**
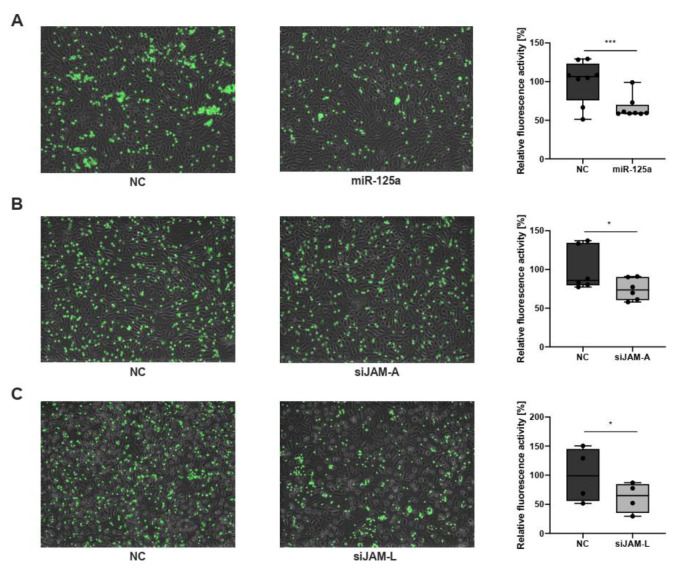
**Overexpression of miR-125a and JAM-A/JAM-L knockdown reduces the adhesion of primary human monocytes to inflammatory endothelial cells**. (**A**) Calcein-stained adherent monocytes transfected with negative control (NC) or miR-125a. Cells were perfused across HUVECs activated with TNF-ɑ. After perfusion and fluorescence microscopy, cells were harvested, and fluorescence intensity was measured (n = 8, *** *p* < 0.001). (**B**,**C**) Calcein-stained adherent monocytes transfected with negative control (NC) or siRNA targeting either JAM-A (n = 6, * *p* < 0.05) or JAM-L (n = 4, * *p* < 0.05).

**Figure 4 ijms-23-10684-f004:**
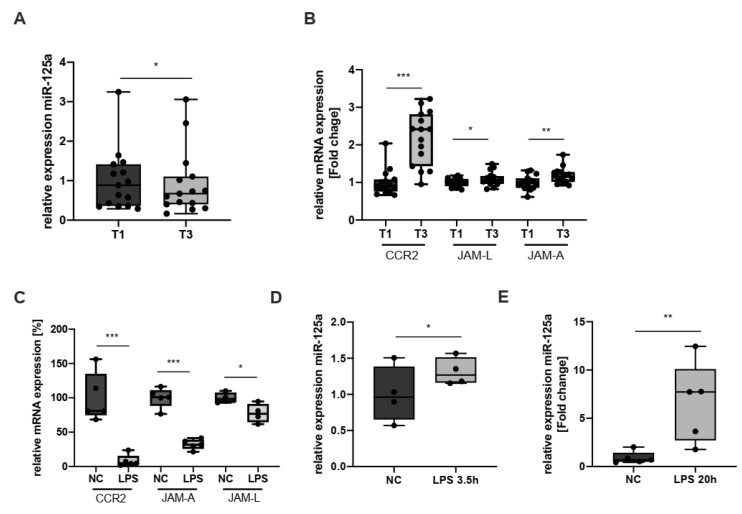
**MiR-125a, CCR2, JAM-A and JAM-L expression in primary human monocytes during the onset of acute inflammation:** (**A**) expression of miR-125a in monocytes isolated from patients before CPB (T1) and on the first postoperative day (T3) (n = 15, * *p* < 0.05); (**B**) CCR2 (n = 15, *** *p* < 0.001), JAM-A (n = 15, *** *p* = 0.001) and JAM-L (n = 15, * *p* < 0.05) levels in monocytes isolated from patients before CPB (T1) and on the first postoperative day (T3); (**C**) mRNA expression of CCR2, JAM-A and JAM-L after incubation with LPS for 3.5 h; (**D**,**E**) miR-125a expression after incubation with LPS for 3.5 (n = 4, *p* < 0.05) or 20 h (n = 5, ** *p* < 0.01).

**Figure 5 ijms-23-10684-f005:**
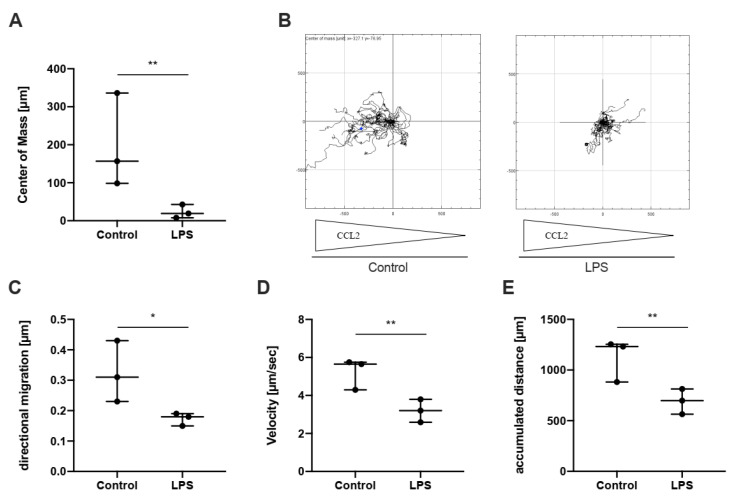
**Effect of TLR-4 activation induced miR-125a on monocyte chemotaxis.** CCR2-specific monocyte chemotaxis recorded by time-lapse microscopy was analysed using single-cell tracking. Control- or LPS-stimulated monocytes were incubated in a chemotaxis slide (ibidi). Arrows below the plot indicate the CCL2 concentration gradient: (**A**) centre of mass represents the average of all single-cell endpoints; (**B**) one representative chemotaxis plot of three independent experiments; (**C**–**E**) directional migration, velocity and accumulated distance in LPS-treated and control monocytes. * *p* < 0.05, ** *p* < 0.01.

## Data Availability

RNA microarray data are available on ArrayExpress under the accession number MTAB-11824. The remaining data are contained within the article or its Supplementary Material.

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
