# Peer review of "A Functional Network Driven by MicroRNA-125a Regulates Monocyte Trafficking in Acute Inflammation"

_ijms, 2022, doi:10.3390/ijms231810684_

Round 1

Reviewer 1 Report

This manuscript analyses the role of miR-125a in regulating monocyte trafficking via several molecules involved in adhesion and trafficking. The results support miR125a regulation of JAM-A, JAM-L and CCR2. The results are derived both in vitro and from patients undergoing cardiac surgery. Stimulation of TLR4 inhibits mir-125a when cells are in inflammed tissue. While of interest there are a few points that that should be addressed;

(1) more detail needs to be given of some results ie fig1 B-E where the text gives % changes but it is not clear to what these refer. Similarly, while 1c shows a change, it is not clear the same effect is seen in 1e.

(2) Fig 2a is not indicated in the text.

(3) is there any further gene changes either via miR-125a or via TLR that changes monocytes into macrophages in tissue or sedentariness. Moreover, if the tissue is still inflammed how would such macrophages help to clear the challenge.

(4) Finally, macrophages are present in most tissues under normal circumstances and in some replaced on a regular basis by monocytes. Would the authors like to comment on whether resident macrophages have a similar phenotype to their TLR stimulated cells, and how monocytes enter under homeostatic conditions.

Author Response

Dear reviewer,

thank you for your comments. Please find our replies below.

(1) In the figure cations of fig 1, we clarified the methods the results were obtained with: the graphs depict the mRNA changes as measured by qRT PCR, the histograms the flow cytometric analysis .

(2) We now mention the illustration that shows the exact localisation of the miR-125a binding sites within the 3`UTRs of the genes of interest in the text.  

(3) For this question, we refer to Banerjee at al (Ref. 46 in the manuscript). Here, the authors nicely show that miR-125a in involved in monocyte-to-macrophage differentiation and guides macrophage polarization. Moreover, we recently showed that monocyte-to-macrophage differentiation (using M-/GM-CSF) in vitro also dramatically alters miR-125a levels. This work was already published by us and is indicated in the mansucript (please see ref. 7)

(4) Resident macrophages within inflamed or non-inflamed tissue show a more or less stationary phenotype. Thus, their migratory and chemotactic abilities are rather low. As monocyte-to.macrophage differentiation dramatically alters gene expression, these cell types, although macrophages arise from moncytes after differentiation- are not comparable at all. Thus, it is difficult to compare inflammatory monocytes to mature resident macrophages. 

Although this mansucript mainly deals with the regulation of monocye trafficking during inflammation, the steady-state trafficking of monocytes to the perivascular area is an important topic of research. Thus, we now discuss physiological monocyte trafficking in Section 2.4

Reviewer 2 Report

The authors evaluate the regulatory mechanisms by which miR-125a directly targets two adhesion molecules, Junction Adhesion Molecule A (JAM-A), Junction Adhesion Molecule-Like (JAM-L), and Chemokine Receptor CCR2, which promotes chemotaxis. They evaluated the regulatory mechanisms that affect monocyte adhesion and chemotaxis by directly targeting two adhesion molecules, JAM-A, Junction Adhesion Molecule-Like (JAM-L), and the chemokine receptor CCR2, which promotes chemotaxis. In monocyte experiments, they also have confirmed that inflammation decreases miR-125a levels and simultaneously enhances the expression of JAM-A, JAM-L, and CCR2. On the other hand, TLR-4-specific stimulation with pathogen-associated molecular pattern (PAMP) LPS dramatically induces miR-125a levels, while simultaneously confirming suppression of JAM-A, JAM-L, and CCR2 expression. I think that their results are valid, clear, and scientifically sound, as are the evaluation methods. I strongly recommend that this paper be published because the results of this study demonstrate the importance of miR125a in the regulatory network and provide important information for understanding the regulatory mechanisms of inflammatory stimuli and for drug discovery.

Comment#

The text in the figure in Fig. 1A is too small to be confirmed

Author Response

Dear reviewer,

thank you for the valuable comment. Accordingly, we magnified the heatmap. However, due to the regulation of numerous genes by miR-125a, the gene names were no readable. Thus, we created clusters and supplied the exact gene names, ordered by clusters as a supplemental information.